# Chaenomeles Fructus (CF), the Fruit of *Chaenomeles sinensis* Alleviates IL-1β Induced Cartilage Degradation in Rat Articular Chondrocytes

**DOI:** 10.3390/ijms23084360

**Published:** 2022-04-14

**Authors:** Changhwan Yeo, Chae Ryeong Ahn, Jai-Eun Kim, Young Woo Kim, Jinbong Park, Kwang Seok Ahn, In Jin Ha, Yoon Jae Lee, Seung Ho Baek, In-Hyuk Ha

**Affiliations:** 1Jaseng Spine and Joint Research Institute, Jaseng Medical Foundation, Seoul 135-896, Korea; duelf12@gmail.com (C.Y.); goodsmile8119@gmail.com (Y.J.L.); 2Department of Science in Korean Medicine, Graduate School, Kyung Hee University, 26 Kyungheedae-ro, Dongdaemun-gu, Seoul 02447, Korea; cofud2917@naver.com; 3College of Korean Medicine, Dongguk University, 32 Dongguk-ro, Ilsandong-gu, Goyang-si 10326, Korea; herbqueen@dongguk.ac.kr (J.-E.K.); ywk@dongguk.ac.kr (Y.W.K.); 4College of Korean Medicine, Kyung Hee University, 24 Kyungheedae-ro, Dongdaemun-gu, Soeul 02447, Korea; thejinbong@khu.ac.kr (J.P.); ksahn@khu.ac.kr (K.S.A.); 5Korean Medicine Clinical Trial Center (K-CTC), Korean Medicine Hospital, Kyung Hee University, Seoul 02447, Korea; ijha0@naver.com

**Keywords:** osteoarthritis, *Chaenomeles Fructus*, *Chaenomeles sinensis*, cartilage degradation, natural product

## Abstract

Osteoarthritis (OA) causes persistent pain, joint dysfunction, and physical disability. It is the most prevalent type of degenerative arthritis, affecting millions of people worldwide. OA is currently treated with a focus on pain relief, inflammation control, and artificial joint surgery. Hence, a therapeutic agent capable of preventing or delaying the progression of OA is needed. OA is strongly associated with the degeneration of the articular cartilage and changes in the ECM, which are primarily associated with a decrease in proteoglycan and collagen. In the progress of articular cartilage degradation, catabolic enzymes, such as matrix metalloproteinases (MMPs), are activated by IL-1β stimulation. Given the tight relationship between IL-1β and ECM (extra-cellular matrix) degradation, this study examined the effects of Chaenomeles Fructus (CF) on IL-1β-induced OA in rat chondrocytes. The CF treatment reduced IL-1β-induced MMP3/13 and ADAMTS-5 production at the mRNA and protein levels. Similarly, CF enhanced col2a and aggrecan accumulation and chondrocyte proliferation. CF inhibited NF-κB (nuclear factor kappa B) activation, nuclear translocation induced by IL-1β, reactive oxygen species (ROS) production, and ERK phosphorylation. CF demonstrated anti-OA and articular regeneration effects on rat chondrocytes, thus, suggesting that CF is a viable and fundamental therapeutic option for OA.

## 1. Introduction

Osteoarthritis (OA) is characterized by the gradual loss and destruction of the articular cartilage, resulting in chronic pain, joint dysfunction, and physical disability. OA is the most common type of degenerative arthritis, impacting millions of individuals worldwide [1]. Increased life expectancy has led to an increase in the incidence of OA [2] and a reduced quality of life among the elderly population [3]. Approximately 303.1 million people worldwide suffer from hip and knee OA [4]. The current treatment for osteoarthritis is focused on pain relief, inflammation suppression, intra-articular injection of hyaluronic acid, and artificial joint surgery, but there are no effective and fundamental treatments [5,6,7].

Recent clinical approaches also include stem cell implantation into knee joints. Such treatments were demonstrated to be partially successful in regenerating cartilage. However, the related mechanisms were unclear, and the treatment is not yet commonly used [8,9,10]. Therefore, an effective therapeutic agent that can control the mechanism of OA progression is urgently needed.

OA is closely related to the loss of articular cartilage, which is mainly composed of proteoglycan, and collagen is degraded through a change in ECM [11]. Catabolic enzymes, when activated, degrade proteoglycan and collagen in the articular cartilage. Matrix metalloproteinase (MMP) plays a key role in articular cartilage destruction. Interleukin-1β (IL-1β), tumor necrosis factor-alpha (TNF-α), and IL-6 are found in the OA cartilage [12]. Among them, IL-1β is linked intimately to ECM degradation [13]. Chondrocytes produce MMP-1, MMP-3, MMP-13, and aggrecanase 1 and 2 (ADAMTS-4 and ADAMTS-5) when stimulated with IL-1β. IL-1β induces NF-κB activation, leading to the expression of ECM degradation products [14]. Therefore, targeting the IL-1β-activated catabolic metabolism could be an effective target in OA treatment.

There are five wild Chaenomeles species in East Asia: *Chaenomeles speciosa*, *Chaenomeles thibetica*, *Chaenomeles cathayensis*, *Chaenomeles sinensis*, and *Chaenomeles japonica* [15]. *Chaenomeles sinensis* is cultivated widely in Korea, and *Chaenomeles Fructus* (CF), the dried fruits of *Chaenomeles sinensis* (commonly known as mugua), has been used in traditional Korean medicine to treat weak muscles, bones, and arthritis [16]. A recent study reported that *Chaenomeles sinensis* exhibits antioxidant, antihyperlipidemic, antitumor, anti-inflammatory, anti-influenza viral, and anti-Parkinson activities [15]. CF is a versatile herb that can be combined with a variety of other medicinal herbs.

Sukjiyanggeuntang, which contains CF among other herbs, has been used to treat musculoskeletal disorders, in particular to treat the symptoms of sciatic crushed nerve injuries [17]. Protocatechuic acid, flavonoids (quercetin and luteolin) and triterpenes (oleanolic acid and ursolic acid) are known as active compounds in Chaenomelis species [18,19]. In addition, those five compounds have been reported to exert anti-OA effects [17,18,19,20,21,22,23,24]. On the other hand, the effects of *Chaenomeles sinensis* on the articular cartilage and OA progression are unclear. This study examined the effects of CF on IL-1β-induced OA progression in rat chondrocytes. NF-κb and MAPKs expression were examined further to understand the mechanism for the delay of ECM degradation.

## 2. Results

### 2.1. UPLC-ESI-QTOF MS/MS Analysis for Identification of Chemical Components in the Water Extract from Chaenomeles Fructus

Ultra-performance liquid chromatography–electrospray ionization/quadrupole-time-of-flight high-definition mass spectrometry/mass spectrometry (UPLC-ESI-QTOF-MS/MS) of the water extract was performed to determine the chemical profile and identify the constituents from the extract. The extracted ion chromatogram shows five known components, which are protocatechuic acid (Retention Time; RT, 3.9 min), quercetin (RT, 9.0 min), luteolin (RT, 9.3 min), oleanolic acid (RT, 25.8 min), and ursolic acid (RT, 26.4 min) of the 10 ppm reference standard individually (Figure 1A).

The five peaks were identified in the water extract of CF (Figure 1B) using UPLC-ESI-QTOF MS/MS in negative ion mode with retention time (RT, min) and mass accuracy (<3 ppm) (Appendix A) and also confirmed with isotope mass patterns and ms/ms spectra (not shown). The chemical profiles of the water extract using LC-MS/MS revealed the presence of thirteen compounds: gallic acid, vanillic acid, protocatechuic acid, neochlorogenic acid, four procyanidin dimer 1–4, catechin, chlorogenic acid, cryptochlorogenic acid, epicatechin, syringic acid, glucopyranosyl-coumaric acid, quercetin, luteolin, and betulinic acid (Appendix A).

### 2.2. CF was Not Cytotoxic to Rat Primary Chondrocytes

The cytotoxicity of CF on rat chondrocytes was assessed using a CCK8 assay. The cell viability of rat chondrocytes was unaffected at concentrations ranging from 0 to 100 μg/mL (Figure 2A). The cytotoxicity of CF ranging from 0 to 200 μg/mL on IL-1β treated rat chondrocytes was also examined. As shown in Figure 2B, CF did not affect the cell viability of rat chondrocytes at concentrations of 0 to 100 μg/mL.

### 2.3. CF Suppressed IL-1β-Induced Cartilage Gene Expression and the Protein Level of MMP3/13 and Adamts5 in Rat Chondrocytes

IL-1β plays a vital role in cartilage degradation and inhibits ECM synthesis by up-regulating major extracellular proteolytic enzymes, such as MMPs and ADAMTS-5 [22,25,26]. The effects of CF on the gene expression of MMP3/13 and ADAMTS5 were determined by co-treating the primary chondrocytes with CF at various concentrations and IL-1β for 24 h. The gene expression levels of *MMP3/13* and *ADAMTS5* were evaluated by real-time PCR. The IL-1β treatment increased the gene expression of *MMP3/13* and *ADAMTS5*, and the CF treatment inhibited the gene expression induced by IL-1β in a dose-dependent manner (Figure 3A). Next, the protein levels were measured using western blot analysis. The protein levels of MMP3/13 and ADAMTS5 also increased in the IL-1β induced chondrocytes, and a co-treatment with CF significantly inhibited the protein levels in a dose-dependent manner (Figure 3B).

### 2.4. CF Prevented IL-1β-Induced Degradation of aggrecan and Col2a1 and Reversed IL-1β-Induced Alcian Blue Staining Loss

Collagen II and aggrecan are the two main components of the cartilage matrix that contribute to the compressive and shock absorption capacity of cartilage [27]. Thus, this study tested the effects of CF on aggrecan and Col2a1 in rat chondrocytes. As shown in Figure 4A, IL-1β stimulation decreased aggrecan and Col2a1 at the mRNA level while these changes were recovered by CF treatment. The recovery effect was dose-dependent. The immunofluorescence assay also shows that CF treatment increased the expression of aggrecan and Col2a1 degraded by IL-1β treatment (Figure 4B,C). In addition, the extracellular matrix was evaluated by Alcian blue staining [28]. As shown in Figure 4D, cartilage degradation induced by IL-1β was restored by the CF treatment.

### 2.5. CF Inhibited IL-1β-Induced NF-κB Activation in Rat Chondrocytes

Previous studies reported that IL-1β-induced NF-κB activation overproduces MMPs, which causes cartilage degradation [22,29]. This study examined the effects of CF on NF-κb activation using western blot analysis. Figure 5A showed that *p*-p65, the activated form of p65 NF-κb, was increased by IL-1β stimulation. CF suppressed the phosphorylation of p65 in a dose-dependent manner. In addition, an immunofluorescence assay was performed to determine whether CF inhibits p65 nuclear translocation. As expected, immunofluorescence analysis showed that the CF (50 μg/mL) treatment prevented the IL-1β-induced p65 translocation to the nucleus (Figure 5B).

### 2.6. CF Repressed IL-1β-Induced ERK2 Activation in Rat Chondrocytes

The activated status of MAPKs (p-ERK, p-JNK, and p-p38) was detected to investigate the effect of CF on the MAPK signaling pathway through western blot analysis. IL-1β stimulation significantly induced activation of the MAPK pathway by increasing the p-p38, p-ERK, and p-JNK levels. Among the pathways, CF inhibited ERK phosphorylation but could not suppress JNK and p38 activation (Figure 6A). Figure 6B shows the western blotting results.

### 2.7. CF Reduced IL-1β-Induced Reactive Oxygen Species (ROS) Production in Rat Chondrocytes

Oxidative stress, which is related to chondrocyte senescence, apoptosis, and extracellular matrix degradation, induces osteoarthritis and synovial inflammation [30]. This study evaluated the effects of CF on ROS production, which is increased by IL-1β stimulation through flow cytometry analysis. As shown in Figure 7 (50 μg/mL) treatment decreased the percentage of cells showing excessive ROS production from 11.4% to 6.8%.

## 3. Discussion

OA is a joint disease with a high prevalence among the elderly and is a huge burden on society. It causes chronic pain and disturbed movement function of the joint [31,32]. A loss of homeostasis in ECM degradation plays a significant role in OA development [33]. MMPs mainly contribute to ECM degradation because of their degradation activity on ECM components, such as aggrecan and collagen II. Increased IL-1β produces MMPs and ADAMTS-4 in OA patients. IL-1β activates the MAP kinase and NF-κB signaling pathways, which then regulate the expression of various genes involved in the synthesis of several inflammatory cytokines and MMPs [34]. ROS also play a critical role in cartilage degradation and chondrocyte death. An excessive level of ROS production may cause oxidative damage on chondrocytes and the activation of the signaling pathways that induce MMPs [35].

Currently, the main purpose of clinical OA treatment is mostly limited to symptom management. Replacement surgeries, which cause long rehabilitation yet result in incomplete recovery, are considered at the last stage in the treatment of OA [36]. Nonsteroidal anti-inflammatory medicines (NSAIDs) are the most frequently prescribed OA medication to alleviate persistent pain and swelling. NSAIDs, however, cannot prevent or delay the degradation of cartilage matter. Furthermore, the long-term use of NSAIDs may result in multiple side effects and drug resistance [36].

Hence, a safe and effective drug that can delay or suppress cartilage degeneration caused by OA is needed. Plant extracts with articular-protective properties and few adverse effects have attracted considerable interest in the treatment of OA [37]. CF is also regarded as a promising therapeutic phytochemical owing to its antioxidant, anti-inflammatory, and anti-Parkinson activities [19]. This study examined the activities of CF on cartilage degradation, NF-κb and MAPK signaling pathways, and ROS production in rat chondrocytes.

First, we confirmed that five compounds were found in CF using UPLC-ESI-QTOF MS/MS in negative ion mode (Figure 1). Quercetin, luteolin, and protocatechuic acid were detected at a higher intensity than oleanolic acid and ursolic acid. In particular, over 15 articles studied the anti-OA effect and joint protective properties of quercetin. Quercetin inhibited oxidative stress-induced apoptosis in rat chondrocytes via SIRT1/AMPK-mediated ER stress suppression and delayed the progression of osteoarthritis in a rat model [38]. Li W at al. found that quercetin ameliorated OA progression by suppressing inflammation and apoptosis in a rat model of OA [39].

Wei B at al. reported that quercetin protected chondrocytes against oxidative stress in a rabbit model of OA [40]. Luteolin and protocatechuic acid were also reported to possess an anti-OA effect [20,22]. In this study, the chemical profile of the CF extract determined by LC-MS/MS (Appendix A) suggests that it contains diverse natural compounds, including polyphenols with antioxidant and anti-inflammatory properties. We verified that CF contains a number of components with anti-OA, anti-inflammatory, and antioxidant properties, and on this basis, we hypothesized that CF may have therapeutic and joint-protecting capabilities.

The CF treatment was safe because the highest concentration of 100 μg/mL did not cause toxicity on chondrocytes (Figure 2). A 50 μg/mL CF treatment prevented IL-1β-induced *MMP3/13* and *ADAMTS-5* overproduction at the mRNA and protein levels (Figure 3). Given that CF treatment decreased adamts5, the primary enzyme responsible for aggrecan degradation [41], we hypothesize that CF treatment inhibited cartilage degradation and increased aggrecan levels.

Next, we determined the level of aggrecan and col2a1 CF that elevated the accumulation of col2a and aggrecan and stimulated chondrocyte proliferation (Figure 4). Furthermore, CF inhibited NF-κb activation and nuclear translocation caused by IL-1β stimulation (Figure 5). IL-1β activated the ERK, JNK, and p38 signaling pathways while CF suppressed ERK phosphorylation (Figure 5), suggesting that inhibition of the NF-κb and ERK signaling pathways is the underlying mechanism for the anti-OA effects of CF.

The CF treatment reduced IL-1β-induced ROS production (Figure 7), which is associated with articular degeneration and senescence. The inflammatory reaction is initiated when IL-1β phosphorylates ERK and p65 within 1 h, and the ROS level increases as a result of the inflammatory reaction, reaching its maximum value after 3 h. It was clearly observed that the level of aggrecan and col2a1 decreased around 24 h because of prolonged inflammatory reactions with increased ROS. However, CF treatment notably recovered this cartilage degradation induced by IL-1β.

In clinical practice, CF is combined with conservative treatments in the form of a medication containing several natural ingredients to treat OA. Since the articular joints do not contain blood vessels, further study is required to prove the efficacy of CF in animal OA models before this can become widely accepted as the primary treatment for OA. However, several studies have suggested that the oral route of the active compounds of CF, such as quercetin, protocatechuic acid, and luteolin, showed positive results regarding the knee joints [20,22,42,43]. Another possible method of administration to maximize the pharmaceutical effect could be local injection.

When quercetin was injected into the knee joint, it was found to have a therapeutic effect [21]. Other components of CF were also proven to be effective with local injection into the knee joint [23,24]. After successfully proving its effect in direct injection into the knee, CF may also be utilized to OA using the local injection method. Our study showed the beneficial effect of CF on OA by showing that it can delay the progression of OA and preserve chondrocytes. These results indicate that CF treatment may be a feasible option for fundamental disease treatment.

## 4. Materials and Methods

### 4.1. Preparation of Extracts from CF Fruit

CF fruit was purchased from Green M.P Pharm. Co. Ltd. (Gyeonggido, Korea). The CF fruit was extracted with distilled water at 105 °C for 3 h and filtered with filter paper. The filtrate was cooled to −20 °C and then lyophilized with a freeze dryer (Ilshin BioBase Co., Ltd., Gyeonggido, Korea). The CF dry extract was stored at −20 °C until use.

### 4.2. Primary Rat Chondrocyte Isolation and Culture

Knee articular cartilage was isolated from three–four-day-old postnatal Sprague–Dawley rats (Orient Bio, Seongnam, Gyeonggi-do, Korea), and digested with 0.2% collagenase type II dissolved in serum-free Dulbecco’s modified Eagle’s medium (Hyclone, Logan, UT, USA) at 37 °C. The debris was filtered in a 40 µm cell strainer, and the chondrocytes were cultured in DMEM containing 10% fetal bovine serum at 37 °C with 5% CO_2_.

### 4.3. Cell Viability

The primary chondrocytes (1 × 10^4^/well) were seeded into a 96-well culture plate for two days and stimulated with IL-1β (10 ng/mL) and different concentrations of CF or IL-1β (10 ng/mL) alone. After 24 h, 10 µL of a Cell Counting Kit-8 (CCK-8; Dojindo, Kumamoto, Japan) solution was added to each well, and the plates were then incubated for 4 h at 37 °C. A microplate reader (Epoch, BioTek, Winooski, VT, USA) was used to assess the cell viability at 450 nm.

### 4.4. Nuclear and Cytoplasmic Extraction

Nuclear and cytosol extraction was performed using NE-PER nuclear and cytoplasmic extraction reagents (Thermo scientific, Waltham, MA, USA) according to the manufacturer’s instruction. Primary chondrocytes (8 × 10^5^/well) were seeded on a 60 mm culture dish. After incubation for two days, the cells were stimulated with IL-1β (10 ng/mL) and different concentrations of CF or IL-1β (10 ng/mL) alone at 37 °C for 1 h. The cells were washed twice with PBS and centrifuged at 500× *g* for 5 min.

The cell pellet was added to cytoplasmic extraction reagent I and incubated on ice for 10 min. The suspension was added to cytoplasmic extraction reagent II and then incubated on ice for 1 min. The suspension was centrifuged at 16,000× *g* for 5 min. The supernatant (cytoplasmic extract) was transferred to a pre-chilled tube. The insoluble fraction (pellet), which contains the nuclei, was re-suspended using a nuclear extraction reagent, incubated for 40 min, and then centrifuged at 16,000× *g* for 10 min. The supernatant (nuclear extract) was used for the experiments.

### 4.5. Western Blot Analysis

Primary chondrocytes (5 × 10^5^/well) were seeded on a six-well culture plate. After incubation for two days, the cells were treated with different concentrations of CF and 10 ng/mL IL-1β at 37 °C for 24 h. The cells were digested in RIPA buffer containing the inhibitors for phosphatases (Millipore, Burlington, MA, USA) and proteases (Millipore, 535140). The total protein lysates (20 µg) were separated by 8% or 10% SDS-PAGE gel and transferred onto a PVDF for 90 min at 100 V.

After blocking with 5% nonfat skim milk for 1 h at room temperature, the membrane was incubated with specific primary antibodies (Table 1) at 4 °C overnight and then specific secondary antibodies for 2 h at room temperature. The proteins were detected using Enhanced chemiluminescence (Bio-Rad, Hercules, CA, USA) and exposed to an Amersham Imager 600 (GE Healthcare Life Sciences, Uppsala, Sweden). The protein expression level was quantified using ImageJ.

### 4.6. Immunofluorescence Assay

Primary chondrocytes (1 × 10^5^/well) were seeded on glass coverslips in a 24 well culture plate. After incubation for two days, the cells were treated with various concentrations of CF with 10 ng/mL IL-1β or 10 ng/mL IL-1β alone at 37 °C for 1 h or 24 h. The cells were fixed with 4% paraformaldehyde for 15 min, kept stable in 0.1% Triton-X in PBS for 15 min, and placed in a blocking buffer (5% bovine serum albumin in PBS) for 1 h at RT.

After washing three times with PBS, the cells were incubated with the primary antibodies (Table 1) overnight at 4 °C and with fluorescein isothiocyanate (FITC)-conjugated secondary antibodies (Jackson Immuno-Research Labs, West Grove, PA, USA) for 2 h at RT. After washing three times with PBS, the nuclei were stained with DAPI (1 µg/mL in PBS) for 10 min at RT. The images were obtained using a confocal microscope (Eclipse C2 Plus, Nikon, Konan, Minato-ku, Japan).

### 4.7. Quantitative Real-Time PCR

Primary chondrocytes (5 × 10^5^/well) were seeded on a six-well culture plate. After incubation for two days, the cells were treated with different concentrations of CF and 10 ng/mL IL-1β at 37 °C for 24 h. The total RNA was extracted from the cells using a TRIzol reagent (Thermo scientific, Waltham, MA USA). The RNA sample was reverse transcribed and synthesized as cDNA using an RT-Kit (Biofact, Daejeon, Korea). Real-time PCR was performed using SYBR Green Master mix (bio-red, 170-8882AP). The samples were normalized to B-actin, the internal control. Table 2 lists the primer sequences used for real-time PCR.

### 4.8. Alcian Blue Stain

Primary chondrocytes were seeded on a 24-well culture plate for 48 h at 37 °C and treated with different concentrations of CF with IL-1β 10 ng/mL or IL-1β 10 ng/mL alone for 48 h at 37 °C. The cells were fixed using 4% paraformaldehyde and then stained with 1% Alcian blue 8 GX in 0.1 N HCL for 2 h at room temperature. Alcian blue was dissolved using 6 M guanidine hydrochloride and measured using a microplate reader at 610 nm.

### 4.9. Flow Cytometry

Primary chondrocytes were seeded on a six-well culture plate for 24 h at 37 °C and then treated with various concentration of CF with IL-1β 10 ng/mL or IL-1β 10 ng/mL alone. After incubation for 3 h, a cell-permeable fluorogenic probe, 2′,7′-dichlorodihydrofluorescein diacetate (DCFDA; Sigma-Aldrich, St. Louis, MO, USA) at 10 μM, was added to the cell pellet for 30 min at 37 °C. The DCFDA fluorescence was measured using a Spectrofluorometer (BD, Franklin Lakes, NJ, USA) at excitation and emission wavelengths of 484 and 530 nm.

### 4.10. Liquid Chromatography–Mass Spectrometry Based Analysis of the Water Extract from Chaenomeles Sinensis Fruit

Chromatographic analysis of the extract was performed to identify the chemical components in the water extract. The extract was shaken with 50% methanol using a vortex mixer for 30 s and sonicated for 10 min. The supernatants were filtered through a 0.2 μm hydrophilic polytetrafluoroethylene syringe filter (Thermo Scientific). The filtrate was then diluted to a 100 mg/mL concentration and transferred to a liquid chromatography (LC) sample vial before use. The liquid chromatography–mass spectrometry system consisted of a Thermo Scientific Vanquish UHPLC system (Thermo Fisher Scientific, Sunnyvale, CA, USA) with a Poreshell EC-C18 column (2.1 mm × 100 mm, 2.7 μm; Agilent, Santa Clara, CA, USA) and a Triple TOF5600+ mass spectrometer system (QTOF MS/MS, SCIEX, Foster City, CA, USA).

The QTOF MS was equipped with an electrospray ionization (ESI) source in negative ion mode and used to complete the high-resolution experiment. The elution program for UHPLC separation used 0.1% formic acid in water as eluent A and methanol as eluent B was as follows: 0–10 min, 5% B; 10–30 min, 5–80% B; 30–31 min, 80–100% B; 31–35 min, 100% B; and equilibration with 5% B for 4 min at a flow rate of 0.3 mL/min. The column temperature was 25 °C, and the auto-sampler was maintained at 4 °C. The injection volume of each sample solution was 5 μL. Data acquisition and processing for qualitative analysis were conducted using Analyst TF 1.7, PeakVeiw2.2 and MasterView (SCIEX, Foster City, CA, USA). The MS/MS data for qualitative analysis were processed using PeakView and MasterView software to screen the probable metabolites based on the accurate mass and isotope distribution.

### 4.11. Statistical Analysis

All experiments were conducted at least three times. The results are reported as the mean ± SD. Statistical analysis between five groups was analyzed by one-way ANOVA followed by a Tukey’s test using a GraphPad Prism software. Differences were considered statistically significant (* *p* < 0.05, ** *p* < 0.01, and *** *p* < 0.001 vs. the control group, # *p* < 0.05, ## *p* < 0.01, and ### *p* < 0.001 vs. the IL-1β treated group).

## Figures and Tables

**Figure 1 ijms-23-04360-f001:**
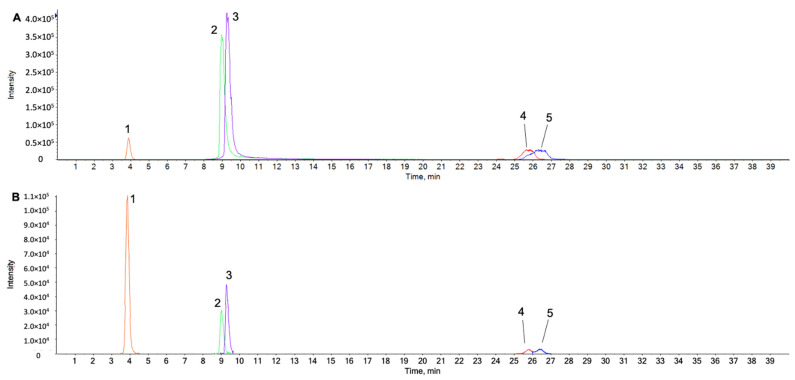
Extracted ion chromatograms (XICs) of five reference standards (**A**) and the five peaks identified in the water extract (**B**) were obtained using LC-ESI-QTOF MS/MS analysis in negative ion mode. Peaks are identified as follows: peak 1; protocatechuic acid (3.86 min), peak 2; quercetin (9.00 min), peak 3; luteolin (9.30 min) peak 4; oleanolic acid (25.82 min), and peak 5; ursolic acid (26.41 min). (Number1)e(number2) represents Number1 × 10^number2^.

**Figure 2 ijms-23-04360-f002:**
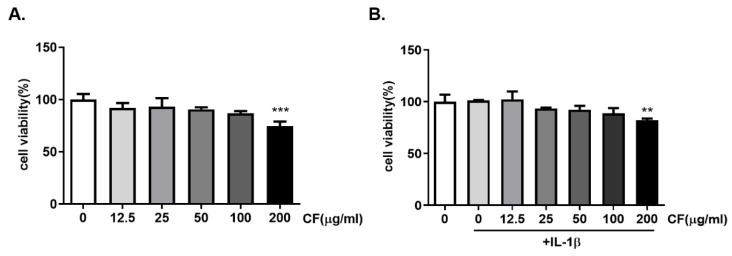
Effects of CF on the viability of primary rat chondrocytes. Primary chondrocytes were incubated with different concentrations of CF (0, 12.5, 25, 50, 100, and 200 μg/mL) for 24 h (**A**). (**B**) shows the effect of IL-1β stimulation on cell viability of rat chondrocytes. The cell viability was assessed using a CCK-8 assay. The data are expressed as the mean ± SD. The results were assessed statistically using one-way ANOVA. ** *p* < 0.01 and *** *p* < 0.001, vs. the control group.

**Figure 3 ijms-23-04360-f003:**
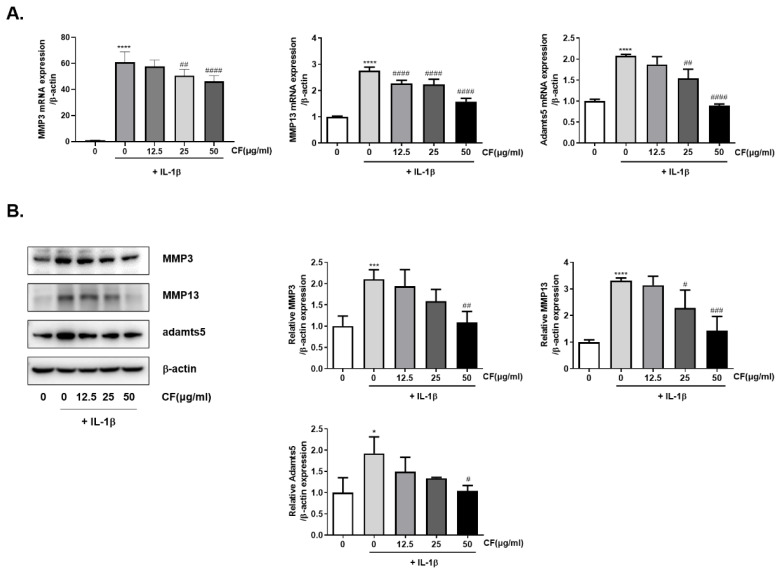
Inhibitory effects of CF on IL-1β induced MMP3, MMP13, and Adamts5 in primary rat chondrocytes. Primary chondrocytes were stimulated with IL-1β (10 ng/mL) with different concentrations of CF (0, 12.5, 25, and 50 μg/mL) for 24 h. (**A**) The mRNA expressions of MMP3, MMP13, and Adamts5 were assessed by real-time PCR. (**B**) The protein expressions of MMP3, MMP13, and Adamts5 were assessed by western blot analysis. * *p* < 0.05, *** *p* < 0.001, and **** *p* < 0.0001 vs. the control group, # *p* < 0.05, ## *p* < 0.01, ### *p* < 0.001, and #### *p* < 0.0001 vs. the IL-1β treated group.

**Figure 4 ijms-23-04360-f004:**
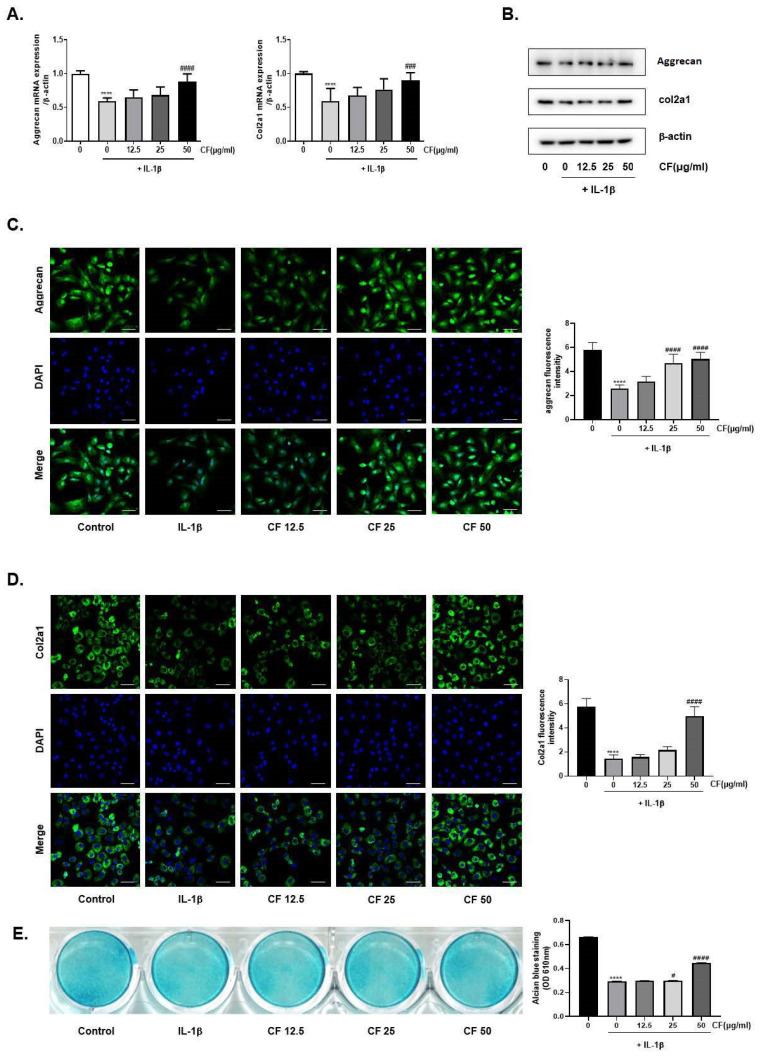
Regeneration effects of CF on IL-1β-induced ECM degradation in primary rat chondrocytes. Primary chondrocytes were stimulated with IL-1β (10 ng/mL) with different CF concentrations (0, 12.5, 25, and 50 μg/mL) for 24 h. (**A**) The mRNA levels of Col2a1 and aggrecan were assessed by real-time PCR. (**B**) The protein expression of col2a1 and aggrecan was assessed by western blot analysis. (**C**) Col2a1 and (**D**) aggrecan were explored by immunofluorescence staining and quantified by image J (Magnification, 400×; scale bar, 50 μm). (**E**) Alcian blue stained with various concentrations of CF for 48 h. Alcian blue was dissolved in 6 M guanidine hydrochloride and measured at 610 nm. **** *p* < 0.0001 vs. the control group, # *p* < 0.05, ### *p* < 0.001, and #### *p* < 0.0001 vs. the IL-1β treated group.

**Figure 5 ijms-23-04360-f005:**
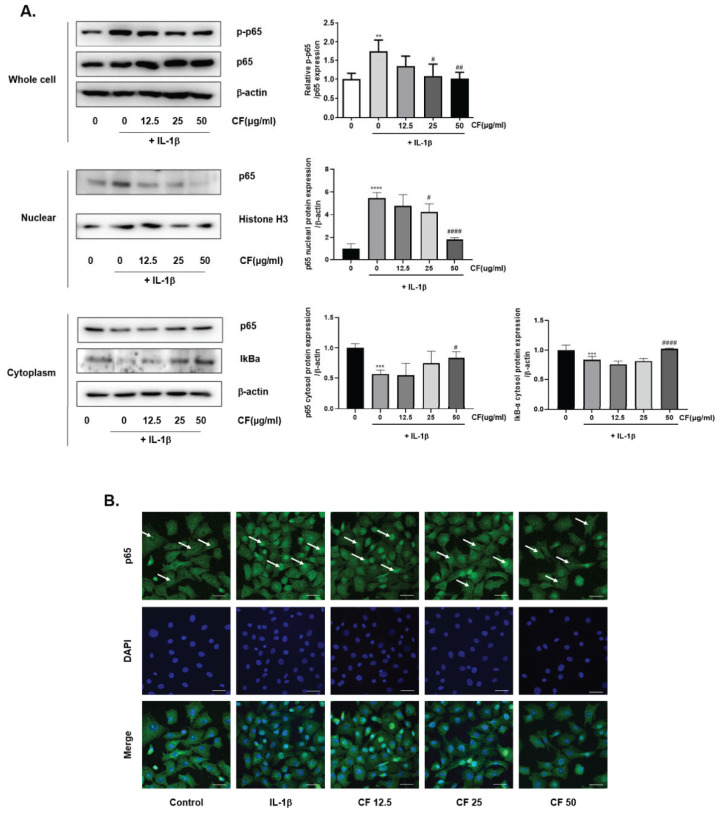
Inhibitory effects of CF on IL-1β-induced NF-κB p65 translocation in primary chondrocytes. Primary chondrocytes were stimulated with IL-1β (10 ng/mL) with different concentrations of CF (0, 12.5, 25, and 50 μg/mL) for 1 h. (**A**) Whole cell, nuclear, and cytoplasmic protein expression of p65 was assessed by western blot analysis. (**B**) p65 translocation into the nucleus was revealed by immunofluorescence staining as tagged by the white arrows (Magnification, 400×; scale bar, 50 μm). ** *p* < 0.01, *** *p* < 0.001, and **** *p* < 0.0001 vs. the control group, # *p* < 0.05, ## *p* < 0.01, and #### *p* < 0.0001 vs. the IL-1β treated group.

**Figure 6 ijms-23-04360-f006:**
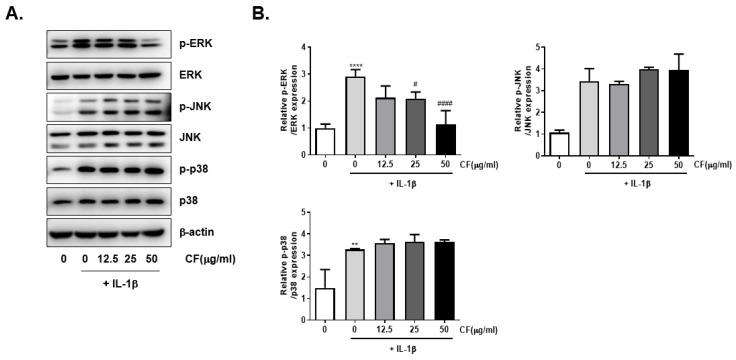
Inhibitory effects of CF on the IL-1β induced MAPK signaling pathway in primary chondrocyte. Primary chondrocytes were stimulated with IL-1β (10 ng/mL) with different concentrations of CF (0, 12.5, 25, and 50 μg/mL) for 30 min. (**A**) The protein expression of ERK, JNK, and p38 was assessed by western blot analysis and (**B**) quantification analysis. ** *p* < 0.01 and **** *p* < 0.0001 vs. the control group, # *p* < 0.05 and #### *p* < 0.0001 vs. the IL-1β treated group.

**Figure 7 ijms-23-04360-f007:**
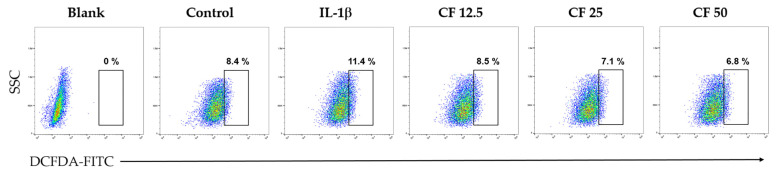
Chondrocyte protection effect of CF on IL-1β-induced ROS generation. Primary chondrocytes were stimulated with IL-1β (10 ng/mL) with different concentrations of CF (0, 12.5, 25, and 50 μg/mL) for 3 h. The ROS levels were measured using flow cytometry.

**Table 1 ijms-23-04360-t001:** List of antibody information.

Antibody	Company	Dilution	Product No.
β-Actin	Santa cruz	1:1000	sc-47778
ADAMTS5	Abcam	1:500	ab41037
MMP-3	Abcam	1:1000	ab53015
MMP-13	Abcam	1:1000	ab39012
COL2A1	Santa Cruz	1:1000	sc-52658
Aggrecan	Abcam	1:1000	ab36861
NF-κB p65	Cell signaling	1:1000	8242 s
p-NF-κB p65	Cell signaling	1:1000	#3033
ERK	Santa cruz	1:1000	sc-81457
p-ERK	Cell signaling	1:2000	#4370
JNK	Cell signaling	1:1000	#9252
P-JNK	Cell signaling	1:2000	#9255
P38	Cell signaling	1:1000	#8690
p-P38	Cell signaling	1:1000	#4511
Histione H3	Cell signaling	1:1000	#9715
IkB-alpha	Cell signaling	1:1000	#9242

**Table 2 ijms-23-04360-t002:** List of primer sequence information.

Primer	Forward	Reverse
MMP3	ATGATGAACGATGGACAGATGA	CATTGGCTGAGTGAAAGAGACC
MMP13	TGCTGCATACGAGCATCCAT	TGTCCTCAAAGTGAACCGCA
COL2A1	TCAACAATGGGAAGGCGTGAG	GTTCACGTACACTGCCCTGAAG
Aggrecan	GCCTCTCAAGCCCTTGTCTG	GATCTCACACAGGTCCCCTC
Adamts5	CAAGTGTGGAGTGTGTGGAG	GTCTTTGGCTTTGAACTGTCG
β-Actin	GCTACAGCTTCACCACCACA	GCCATCTCTTGCTCGAAGTC

## Data Availability

Not applicable.

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
