# Peer review of "Chaenomeles Fructus (CF), the Fruit of Chaenomeles sinensis Alleviates IL-1β Induced Cartilage Degradation in Rat Articular Chondrocytes"

_ijms, 2022, doi:10.3390/ijms23084360_

Round 1
Reviewer 1 Report
The study is devoted to the in vitro evaluation of natural extract (Chaenomeles Fructus) potential in osteoarthritis prevention and healing. The search of drugs and medicines that target OA is an up to date task that will contribute to the improvement of active life duration and overall quality. Cultured rat chondrocytes are used as a model. Degradation is induced by treatment with IL-1β. The methods include cell viability and Western blot analyses, immunofluorescence assay, quantitative real-time PCR, histochemistry. Additionally, fruit extract is analyzed by the means of chromatography and mass spectrometry. The study aims are clearly stated and methodology is well designed. The results show that CF has inhibitory effect on expression of cartilage proteases thus decreasing the IL-1β-induced degradation of matrix components. The possible biochemical mechanisms of CF action via its effect on signaling pathways and ROS production are proposed. The obtained results are clearly presented, illustrated and described. However, the comprehensive analysis of the results is lacking in Discussion. Access for supplemented materials has not been provided. The manuscript can be accepted for publication after addressing the following comments and questions.
General comments
Introduction: the modern methods of OA treatment, such as stem cells implantation, should be mentioned in the literature review.
Materials and Methods. (1) 4.2: the particular sites of articular cartilage extraction should be pointed out. (2) The non-commonly used methods abbreviations should be given in full when first mentioned, for example: RT, ECL, FITC, LC, etc. (3) 4.5: manufacturer of confocal microscope should be outlined.
Results: (1) Fig.2A: the quantity of gene expression related to MMP3 decreased from 0 to 25 µg/ml CF and then again increased for 50 µg/ml CF – this effect should be given an explanation.
(2) Fig.3B: add the bar length in immunofluorescence images.
(3) Fig.3C: in the left image the text (concentrations) does not correspond to the images, please, correct.
(4) Fig.4B: please, indicate the bar length for immunofluorescence images.
(6) Fig.6A (is there Fig.6B?): the axes text is too small and can hardly be read, please, improve. The axes names and particular units should be given for clarity.
In general, the content of the Discussion section duplicates the Results section and does not provide deep analysis of the obtained results. The meaningful critique analysis of the results in relation to the known literature has to be done in Discussion. The significance and limitations of the work should be clearly stated. The following problems should be regarded: (1) What follows from the chemical analysis of CF? What substance (or their groups) may be responsible for the observed curative effects? (2) What is the basis for experimental optimization of cell pretreatment time for various types of analysis: 24 h for Aggrecan and Col2a1, 1 h for NF-kB p65, 30 min for MAPK, 3 h for ROS? (3) What is the authors’ prognosis for translation of this technique to in vivo applications? How can CF be delivered to chondrocytes through dense tissue matrix which does not contain blood vessels? (4) What are the study limitations?
Supplemented materials should be provided for review.
Minor comments
Lines 5-6: check for the upper indices
Lines 46-47: fragment needs English editing
Line 168: false reference to Fig.1B. Figs 7A and 7B should be mentioned in the main text.
Author Response
We sincerely appreciate the effort the reviewer has put on our manuscript. The attached file contains our responses. The critical comments have significantly improved our review. We hope our revised manuscript now meets the satisfaction of the reviewer.

Reviewer 2 Report
In the current manuscript, the authors show an OA therapeutic effect for CF natural extract in chondrocytes cultured with IL-1b in vitro, by reducing the production of proteases, increasing the accumulation of cartilage matrix proteins, and inhibition of NF-kB/ ERK MAPK and ROS activation. The authors present their results nicely, however, there are few points that need to be addressed for a better presentation of this study.
- In Figure 3B, western blotting will provide quantitative evidence, instead of IF.
- In Figure 3, it is necessary to quantify the protein expression of Aggrecan-Neo or NIETGE to determine whether higher levels of Col II and aggrecan are due to increased production or reduced degradation.
- In Figure 3C, Alcian blue is a better technique to measure cartilage matrix production in vitro, instead of Saf O.
- The authors propose NF-kB activation and nuclear translocation as a mechanism of action of CF. However, IF staining is not sufficient to support this conclusion. Cell fractionation and WB of p65 in the nuclear fraction is needed.
- Finally, please discuss the findings in relation to the literature, rather than providing a short description of the pathways then presenting the results. It is important to correlate these findings with in vivo OA models or clinical OA.

Author Response

(The authors gave the same response as above.)

Reviewer 3 Report
This manuscript by Yeo et al. explores the effects of Chaenomeles Fructus (CF) on IL-1β stimulated cartilage degeneration in primary rat chondrocytes. The authors report that CF is able to reduce the mRNA and protein levels of OA proteolytic enzymes by a dose-dependent manner while inducing the production of Col2 and sulphated proteoglycans, possibly through inhibition of nuclear translocation of NF-κB p65.
The work is interesting, and the authors used a variety of experimental approaches to obtain the results and draw conclusions. However, after careful consideration, some points need to be addressed:
- The introduction does not provide sufficient information for CF and the active substances that contain and their role in OA.
- Legends should be edited. For example, there is no statistical analysis with # as a symbol used in Fig1.
- The authors report that CF inhibits the gene expression of MMP3/13 and ADAMTS5 in a dose-dependent manner. However, this is not evident in Fig2A. In addition, MMP-13 cannot be quantified with the Western blot shown in Fig2B. Please comment.
- Lines 110-111, “the extracellular matrix was evaluated by safranin O staining”. This should be rephrased.
- Can IF results for Col2a1 and Acan be quantified in Fig3B?
- Quantification of SafO in Fig3C seems to be problematic since there is unspecific binding.
- NF-kB p65 translocation results based on Fig4B are inconclusive. Higher magnification and use of arrows would be helpful. Scale bar value is missing.
- It is not clear how ROS production was assessed and this part is missing from the methods section.
- Is Fig7A an overlay of different runs using standards? This is not mentioned as well as the concentration of the standards. Which is the wavelength? The same stands for Fig7B, for example, peaks 2 and 3 do not seem to derive from the same chromatogram and they are not separated. The colours used in this Figure lead to confusion. This section should be placed first in the results. Finally, FigS1 and Table S1 cannot be found by the reviewer.
- Discussion is not well written and is a repetition of the results.
Author Response

(The authors gave the same response as above.)

Round 2
Reviewer 1 Report
The manuscript can be accepted for publication after minor text editing. Please, insert spacing between numbers and units (for example, 610 nm, 30 min, etc.)
Author Response
We sincerely appreciate the effort the reviewer has put on our manuscript. Spacing(nm, min, %, etc.) has been corrected.
Reviewer 2 Report
The authors performed all the experiments we requested in the first review. However, the western blots in Figure 5A need quantification, please complete the quantification to confirm the results on which the discussion and conclusions are based.

Author Response
We have performed additional western blotting and quantified Figure 5A.
We sincerely appreciate the effort the reviewer has put on our manuscript.
Reviewer 3 Report
The authors sufficiently addressed my comments.
Author Response
We sincerely appreciate the effort the reviewer has put on our manuscript.